# Implication of m6A mRNA Methylation in Susceptibility to Inflammatory Bowel Disease

**Maialen Sebastian-delaCruz** [1,2], **Ane Olazagoitia-Garmendia** [1,2], **Itziar Gonzalez-Moro** [2,3], **Izortze Santin** [2,3,4], **Koldo Garcia-Etxebarria** [5] and **Ainara Castellanos-Rubio** [1,2,4,6,*]

1   Department of Genetics, Physical Anthropology and Animal Fisiology, University of the Basque Country, 48940 Leioa, Spain; maialen.sebastian@ehu.eus (M.S.-d.); ane.olazagoitia@ehu.eus (A.O.-G.)
2   Biocruces Bizkaia Health Research Institute, 48903 Barakaldo, Spain; itziar.gonzalezm@ehu.eus (I.G.-M.); izortze.santin@ehu.eus (I.S.)
3   Department of Biochemistry and Molecular Biology, University of the Basque Country, 48940 Leioa, Spain
4   CIBER (Centro de Investigación Biomédica en Red) de Diabetes y Enfermedades Metabólicas Asociadas (CIBERDEM), Instituto de Salud Carlos III, 28029 Madrid, Spain
5   Hepatic and Gastrointestinal Disease Area, IIS Biodonostia, 20014 Donostia, Spain; koldo.garcia@biodonostia.org
6   Ikerbasque, Basque Foundation for Science, 48013 Bilbao, Spain
*   Correspondence: ainara.castellanos@ehu.eus

**Abstract:** Inflammatory bowel disease (IBD) is a chronic inflammatory condition of the gastrointestinal tract that develops due to the interaction between genetic and environmental factors. More than 160 loci have been associated with IBD, but the functional implication of many of the associated genes remains unclear. N6-Methyladenosine (m6A) is the most abundant internal modification in mRNA. m6A methylation regulates many aspects of mRNA metabolism, playing important roles in the development of several pathologies. Interestingly, SNPs located near or within m6A motifs have been proposed as possible contributors to disease pathogenesis. We hypothesized that certain IBD-associated SNPs could regulate the function of genes involved in IBD development via m6A-dependent mechanisms. We used online available GWAS, m6A and transcriptome data to find differentially expressed genes that harbored m6A-SNPs associated with IBD. Our analysis resulted in five candidate genes corresponding to two of the major IBD subtypes: *UBE2L3* and *SLC22A4* for Crohn's Disease and *TCF19*, *C6orf47* and *SNAPC4* for Ulcerative Colitis. Further analysis using in silico predictions and co-expression analyses in combination with in vitro functional studies showed that our candidate genes seem to be regulated by m6A-dependent mechanisms. These findings provide the first indication of the implication of RNA methylation events in IBD pathogenesis.

**Keywords:** inflammatory bowel disease; Crohn's disease; ulcerative colitis; m6A; SNP; METTL3; YTHDF1; inflammation

## 1. Introduction

Inflammatory bowel disease (IBD) is a common chronic inflammatory gastrointestinal disorder whose major subtypes are Crohn's disease (CD) and ulcerative colitis (UC) [1]. Ulcerative colitis is characterized by an inflammation limited to the colon, while Crohn's disease involves any part of the gastrointestinal tract, generally the terminal ileum or the perianal region. The general believe is that these diseases develop due to a continuous and inappropriate inflammatory response to intestinal microbes and foreign antigens in genetically susceptible individuals [2–4]. However, the exact mechanisms by which they are triggered remain mostly unknown. Although anti-TNF agents

have been widely used for the treatment of IBD, new treatment options are being actively explored, mainly due to the loss of response to anti-TNF therapy observed in around 30% of IBD patients [5].

Meta-analyses of multiple GWAS have implicated in more than 160 genetic loci in IBD susceptibility, both common and specific for each subtype [6]. Functional analyses of the associated genes have revealed multiple pathophysiological mechanisms related to these variants. In this sense, the CARD15/NOD2 region has been related to an impaired antimicrobial defense and the *TNFSF15* gene has been involved in the induction of the proinflammatory environment, which is a characteristic feature of the disease [7]. Nevertheless, although some of the risk regions have been functionally characterized, the implication of the majority of these loci in the development of the disease are yet to be determined. In trying to elucidate the mechanisms involved in IBD development, some studies have shown that epigenetic changes, such as DNA methylation, could be also involved in the susceptibility of this disease. Trying to elucidate the mechanisms involved in IBD development, some studies have shown that epigenetic changes, such as DNA methylation, could be also involved in the susceptibility of this disease [8].

N6-methyladenosine (m6A) modification is the most abundant internal chemical modification in mRNAs. This dynamic process is implicated in multiple aspects of RNA metabolism, representing a novel component of genetic regulation termed epitranscriptomics. Nowadays we know that m6A marks regulate the stability and structure of RNA molecules, which can, in turn, regulate gene expression [9,10]. SNPs located within or near m6A motifs (m6A-SNPs) have been suggested as potential contributors to disease pathogenesis and different databases of genetic variants related to m6A modifications have been developed in the last few years (e.g., m6Avar and m6ANSP [11,12]).

Proteins that regulate m6A RNA modifications, such as METTL3, WTAP or FTO, have been involved in the development of different diseases due to their ability to alter m6A RNA modification levels [13,14]. Indeed, it has been speculated that m6A RNA methylation can be involved in immune tolerance by regulating the activity of immune pathways and by controlling the development of T lymphocytes [15,16]. In addition, m6A related changes have also been associated with alterations in gut microbiota and with the development of gastrointestinal cancers [17,18]. However, the implication of disease associated SNPs in mRNA methylation and its relationship with IBD pathogenesis remains unexplored.

In this study, we have systematically scrutinized online available GWAS, m6A and transcriptome data to find IBD associated genes putatively regulated by m6A related mechanisms. Moreover, we have observed that IFNγ, which is a distinctive inflammatory feature of IBD, induces m6A methylation in intestinal cells. Further studies are needed to clarify the real impact of disease-associated variants in m6A RNA modifications and their implication in IBD pathogenesis.

## 2. Results

### 2.1. IBD Associated SNPs Are Located Near m6A Motifs in Differentially Expressed Genes

In order to find IBD associated SNPs that could be involved in disease pathogenesis via m6A related processes (m6A-SNPs), we crossed GWAS data from IBD [19] with m6A peak datasets of two types of epithelial cells (two HeLa sets and one HepG2 set) (Figure 1A). The combination of the 6245 IBD, 4973 CD and 2234 UC associated SNPs with the available 21,488, 20,489 and 19,535 m6A motifs from epithelial cells resulted in 446 matches in IBD, 310 in CD, and 81 in CD (Figure 1B, left). Only m6A-SNPs detected in the three m6A datasets were considered for further analyses. Thus, we identified 55 m6A-SNPs in IBD, 36 in CD, and 13 in UC—nine of them were detected in both IBD and its subtypes (Figure 1B, right). These m6A-SNPs correspond to a total of 22 genes associated to IBD, 12 genes associated to CD, and 6 genes associated to UC (Figure 1C). Interestingly, m6Avar tool did not predict any of these genes, while the m6ASNP tool only predicted one of them. Thus, this strategy allowed us to build a new dataset of IBD associated m6A-SNPs and candidate genes.

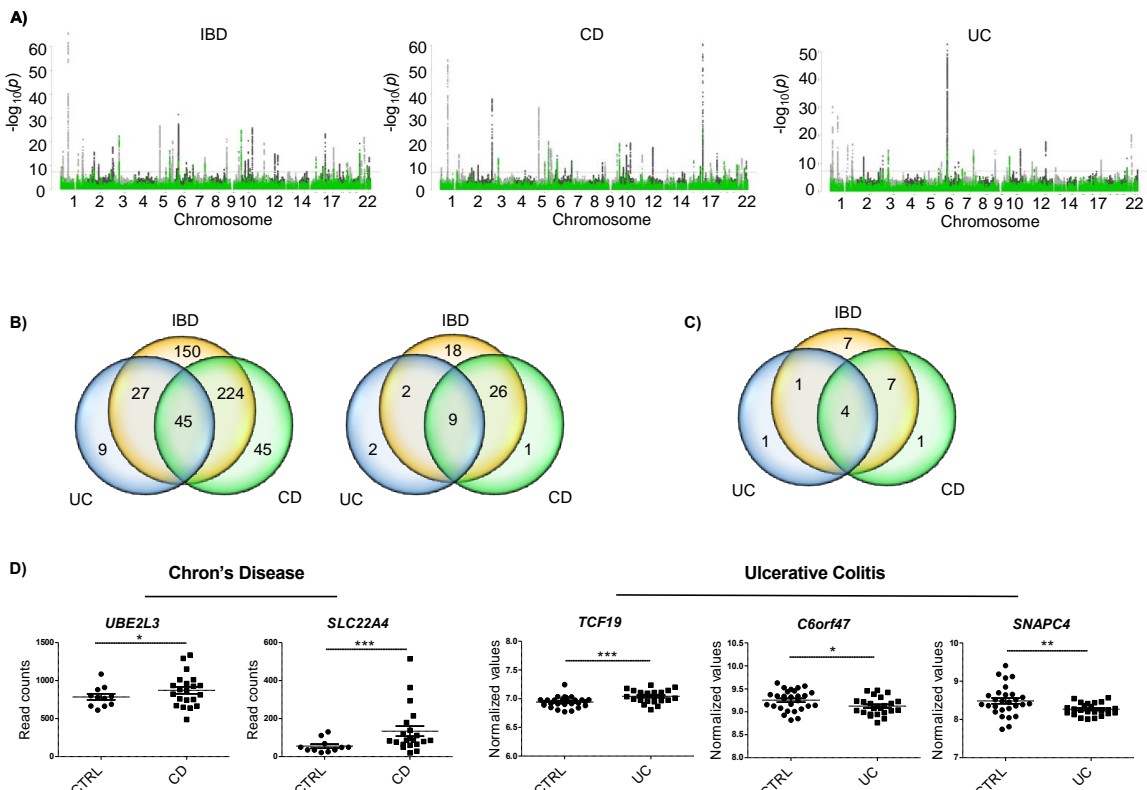

**Figure 1.** (**A**) Manhattan plots of IBD, CD and UC GWAS results; variants that overlap m6A peaks in HeLa and HepG2 cells are in green. Red line depicts the genome-wide significance level ($p < 5 \times 10^{-8}$). (**B**) Venn diagrams showing the amounts of m6A-SNPs in HeLa and/or HepG2 cells (left) and those m6A-SNPs that were common for the three m6A datasets analyzed (right). (**C**) Number of genes harboring m6A-SNPs. (**D**) Differentially expressed genes that harbored a m6A-SNP in each disease subtype. Whole genome expression data for each disease subtype were downloaded from GEO * $p < 0.05$; ** $p < 0.01$; *** $p < 0.001$.

In order to narrow down the list of candidates, whole genome expression data of CD and UC were analyzed. A total of 12 genes with m6A-SNPs were found to be differentially expressed (Supplementary Figure S1). Our aim was to identify m6A-related genes specific for IBD subtypes, hence genes harboring an m6A-SNP and showing differential expression in the subtype that the m6A-SNP was associated to were selected. This selection resulted in five candidates: two genes for CD (*UBE2L3* and *SLC22A4*) and three genes for UC (*TCF19*, *C6orf47* and *SNAPC4*) (Supplementary Table S1).

*UBE2L3* and *SLC22A4* genes showed upregulated expression in CD cases compared to controls. In the case of UC, *TCF19* was found to be upregulated, and *C6orf47* and *SNAPC4* were downregulated in patients compared to the controls (Figure 1D).

## 2.2. m6A SNPs Can Affect Different Layers of RNA Regulation

To assess the relation between the potential m6A-SNPs and the differential expression of candidate genes, we analyzed whether the associated SNPs were influencing the expression of the candidate genes in the disease target tissue (e.g., colon). We found that m6A-SNPs in CD candidate genes (*UBE2L3* and *SLC22A4)* were significant *cis*-eQTLs in the colon (Figure 2A).

In order to find out other mechanisms underlying the regulation of candidate gene expression, we examined whether any RNA binding protein (RBP) was directly binding any of the m6A-SNP regions. We found that the YTHDF2 and FAM120A proteins bound to SNP rs7445, located in the *UBE2L3* gene. Additionally, YTHDF2 also binds to rs148844907 in *C6orf47*. FAM120A participates in the cytosolic transport of the mRNA and has been involved in the development of gastric cancer [20].

In turn, YTHDF2 is an m6A reader protein that affects RNA stability by promoting the degradation of m6A containing mRNA transcripts [21], and thus, the fact that it was bound to those candidate genes suggested a methylated status of those loci.

As the RNA secondary structure of both UTRs and gene bodies has been described to be important in different aspects of mRNA regulation, we also analyzed the possible effects of m6A-SNPs in the secondary structure of our candidate genes. We found that the SNP rs7445 located in the 3'UTR of *UBE2L3* was predicted to change the secondary structure of the mRNA molecule (Figure 2B). Differences in secondary structure can change the accessibility of the m6A marks in 3'UTRs, affecting the stability of the mRNA [22] which could explain the differential expression of *UBE2L3* observed in UC. Additionally, synonymous SNP rs3812565 in the *SNAPC4* gene was also predicted to affect the secondary structure of the mRNA (Figure 2C), which could affect exon recognition, subsequently altering splicing. All the findings regarding regulation are summarized in Table 1.

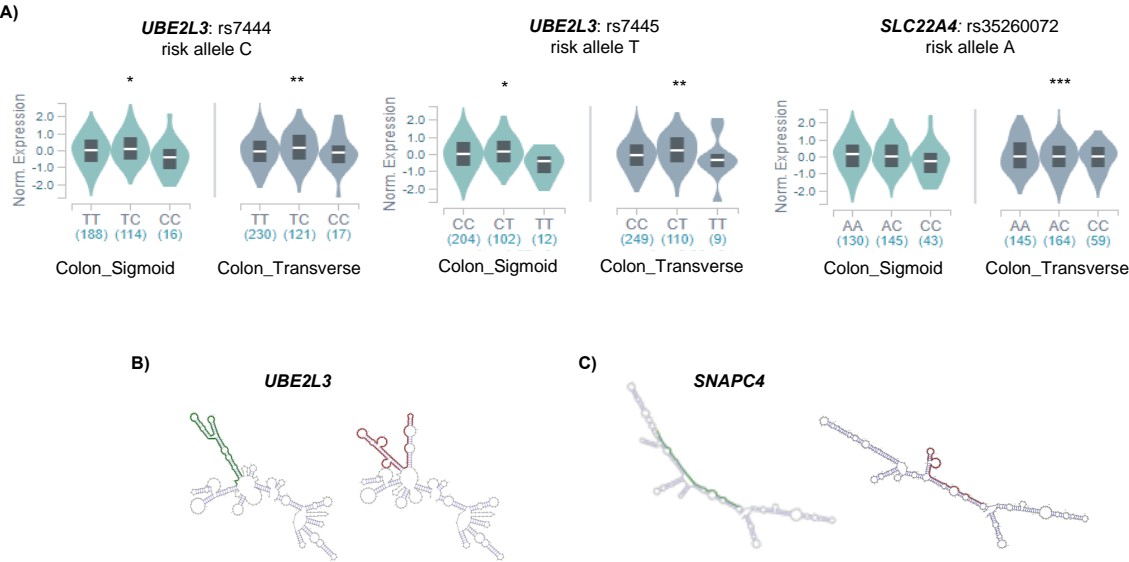

**Figure 2.** (**A**) Significant *cis*-eQTLs found in the colon using GTEX data; * $p < 0.05$; ** $p < 0.01$; *** $p < 0.001$. Allele specific RNA secondary structures as predicted by RTH RNAsnp Web Server tool for (**B**) *UBE2L3* and (**C**) *SNAPC4*. The optimal secondary structures of the global RNA sequences are shown, and the SNP flanking regions of the protective and risk alleles are represented in green and red, respectively.

**Table 1.** Functional predictions of m6A-SNPs within the candidate genes. CS: coding sequence; ns: not significant.

| Gene | Associated SNP | Disease | SNP Location | Functional Predictions | | | | | In Vitro Effect | |
|------|----------------|---------|--------------|-----------|------|------------|-------------------------|-----------|----------|----------|
| | | | | Structure | eQTL | Expression | m6A Binding Proteins | Other RBP | ovMETTL3 | siYTHDF1 |
| *UBE2L3* | rs7444, rs7445 | CD | 3'UTR | YES | YES | CD, up | YTHDF1, YTHDF2 | FAM120A | ns | ns |
| *SLC22A4* | rs35260072 | CD | Intron | Not observed | YES | CD, up | WTAP, METTL3, METTL14, YTHDF1 | No | ns | ns |
| *TCF19* | rs139102013 | UC | Intron | Not observed | ns | UC, up | YTHDF1 | No | ns | ns |
| *C6orf47* | rs148844907 | UC/CD | 5'UTR | NO | ns | UC, down | YTHDF1, YTHDF2 | No | Yes, down | ns |
| *SNAPC4* | rs3812565 | UC/CD | CS, Synonimous | NO | ns | UC, down | WTAP, METTL3, METTL14, YTHDF1 | No | ns | Yes, up |

### 2.3. Genes Harboring m6A-SNPs Interact and Co-Express with m6A Machinery Proteins

In order to analyze which members of the m6A machinery were involved in the regulation of IBD candidate genes, the interaction between m6A machinery proteins and the genes harboring m6A-SNPs was analyzed using the TREW (Target of Reader, Eraser and Writer) database in the two epithelial cell lines used for m6A peak selection. We found that the YTHDF1 reader protein was predicted to interact with all five genes. Additionally, METTL3, METTL14 and WTAP writers also showed interaction with the *SLC22A4* and *SNAPC4* genes (Table 1). Previous interactions with YTHDF2 could not be confirmed using this database, as this reader protein is not among the machinery proteins compiled.

The expression of the m6A machinery proteins that interact with the candidate genes was assessed using the whole genome expression data from CD and UC patients [23,24]. No changes in m6A proteins were observed in CD. However, the *YTHDF1* reader was downregulated while the *WTAP* writer was upregulated in UC patients compared to the control individuals (Figure 3A).

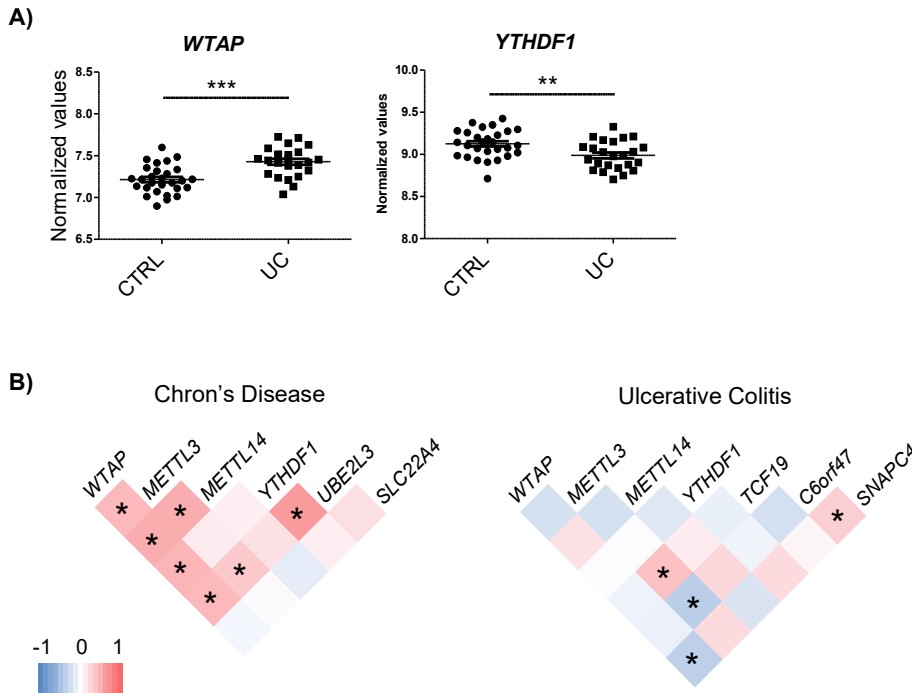

**Figure 3.** (**A**) Differential expression of m6A machinery proteins in UC. (**B**) Co-expression analyses of m6A machinery proteins and candidate genes in each disease subtype. Analysis was performed by Pearson correlation; * $p < 0.05$.

To assess whether changes in the m6A machinery could regulate the expression of the candidate genes, a co-expression analysis was carried out. Although general expression changes were not observed in the m6A proteins, co-expression analysis pointed to the implication of the m6A machinery in the regulation of the expression of target genes. In the CD dataset, *WTAP*, *METTL3* and *YTHDF1* showed positive correlation with *UBE2L3*, suggesting that an increase in m6A could be involved in the induction of this gene. In the UC dataset, *METTL3* expression correlated with *TCF19*, *C6orf47*, and *WTAP* with *SNAPC4* (Figure 3B). Intriguingly, we observed that certain m6A regulatory proteins that co-expressed with candidate genes had not been described to bind these targets and some of the m6A proteins that have been found to interact with candidate genes did not show any co-expression. Only YTHDF1–*UBE2L3* and WTAP–*SNAPC4* pairs co-expressed and were predicted to interact. For the rest of the pairs, the underlying regulation mechanism could be more complex than m6A machinery induction, resulting in candidate gene expression alterations.

## 2.4. IBD-Related Inflammatory Conditions Alter m6A-Related Processes

Cytokines are considered major drivers of the excessive immune response observed in IBD patients, IFNγ being one of the most highly upregulated cytokines in the intestinal mucosa from patients [25]. Given that we observed some changes in mRNA expression on m6A machinery members in patients with UC (Figure 3A), we stimulated HCT116 intestinal cells with IFNγ in order to see if overall m6A levels were influenced by the presence of this cytokine. We observed that total m6A levels were increased when the cells were exposed to IFNγ (Figure 4A). Members of the m6A machinery also showed changes in their expression—m6A reader *YTHDF1* and the writers *METTL3* and *METTL14* were significantly augmented after IFNγ stimulation (Figure 4B). The increase in YTHDF1 and METTL3 in response to IFNγ was also confirmed at the protein level (Figure 4C). Analysis of candidate genes did not show statistically significant changes at mRNA level after 4 h of IFNγ stimulation (Supplementary Figure S2).

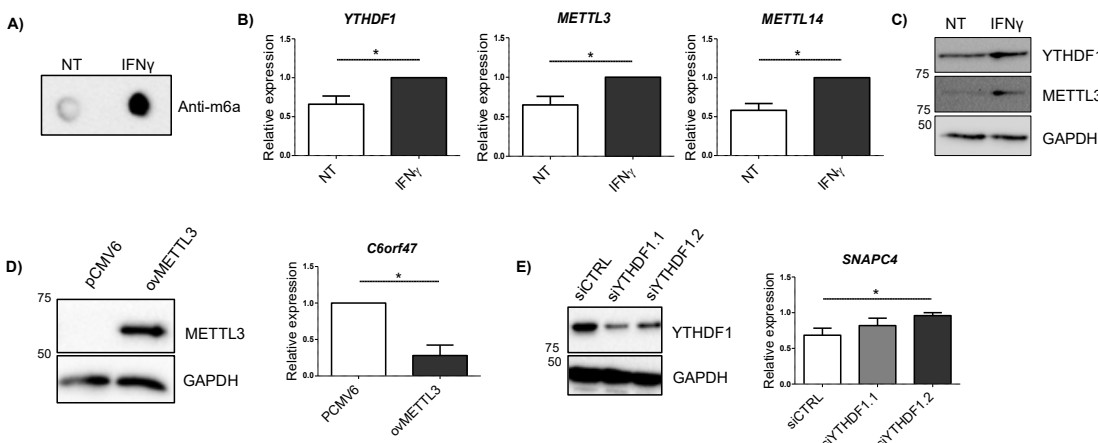

**Figure 4.** (**A**) m6A dot blot of untreated cells (NT) and cells treated with IFNγ for 4 h (IFNγ). Blot is representative of three independent experiments. (**B**) Relative expression of m6A machinery proteins differentially expressed after 4 h IFNγ stimulation. NT: untreated cells; IFNγ: cells treated with IFNγ for 4 h. Data are represented as the mean and standard error of four independent experiments. * $p <$ 0.05 by two tailed Students *t*-test. (**C**) YTHDF1 and METTL3 immunoblot of untreated cells (NT) and cells treated with IFNγ for 4 h (IFNγ). GAPDH was used as a loading control. Blot is representative of three independent experiments. (**D**) METTL3 immunoblot for cells transfected with an empty vector (pCMV6) and a METTL3 overexpressing vector (ovMETTL3). GAPDH was used as a loading control (left). Expression of *C6orf47* in cells overexpressing METTL3 compared to empty vector (right). Data represent mean and standard error of three independent experiments * $p <$ 0.05 by Student's *t*-test. (**E**) YTHDF1 immunoblot for cells transfected with an siRNA control (siCTRL) and two siRNAs targeting YTHDF1 (siYTHDF1.1 and siYTHDF1.2). GAPDH was used as a loading control (left). Expression of *SNAPC4* in cells transfected with a siRNA control or siRNAs targeting YTHDF1 (right). Data represent mean and standard error of three independent experiments * $p <$ 0.05 by Student's *t*-test.

## 2.5. Alterations in METTL3 and YTHDF1 Affect Candidate Gene Expression

In order to confirm if alterations induced by IFNγ in METTL3 and YTHDF1 proteins could regulate candidate gene expression, we modified the expression of these m6A related proteins in intestinal cells. The overexpression of METTL3 resulted in an increase in *C6orf47* (Figure 4D), resembling what was observed in UC patients (Figure 1D) and confirming the involvement of m6A methylation in the regulation of this gene. Additionally, the silencing of YTHDF1 showed an increased expression of *SNAPC4* (Figure 4E). Although no significant changes were observed in other candidate genes when silencing YTHDF1, some candidate genes showed a trend towards upregulation (Supplementary Figure S3). As the YTHDF1 reader is implicated in translation rather than in mRNA expression [22],

the assessment of protein level amounts could give further information about the regulation of these candidate genes by this m6A reader.

## 3. Discussion

m6A RNA modifications are gaining interest due to their implication on a wide range of biological processes. Recent studies have described the involvement of these modifications in immune and inflammatory responses [26,27], but their implication in the development of inflammatory bowel diseases has not been evaluated so far. As the functional implication of the vast majority of the IBD-associated SNPs is still unknown, we have taken advantage of online available GWAS, m6A and transcriptome data to select IBD candidate genes that could be regulated by m6A related mechanisms. Additionally, we also performed some in vitro experiments in intestinal cell lines to determine the potential implication of m6A in IBD pathogenesis.

Although different databases have been developed to find SNPs that could be affecting m6A methylation levels [11,12], the different methods to describe m6A motifs and the huge amount of available GWAS data make it complicated to get unified results. Since m6A marks are described to be tissue specific, we decided to select m6A peaks of two different epithelial cell types, as they are the main cells of IBD target tissue. Methylated sites were crossed with IBD associated SNPs, and genes harboring m6A-SNPs that were differentially expressed in Crohn's Disease or Ulcerative Colitis were selected for further analysis. The list of m6A-SNPs we got using this approach did not overlap with the output of any of the available databases, highlighting the importance of the data origin in this type of analysis. Although, selecting differentially expressed genes reduced the list of the candidate genes to a more manageable number, it is worth mentioning that we found more than 50 m6A-SNPs that were not located into differentially expressed genes, but that could also be involved in disease pathogenesis mechanisms independent of gene expression.

We analyzed whether the m6A-SNPs were directly influencing gene expression using an eQTL approach, but we observed that the expressions of only two out of five candidates were affected by the associated SNP genotype. Moreover, *UBE2L3* was found to be overexpressed in CD while the risk allele was associated with lower expression of this gene, confirming that the relationship between m6A-SNPs and gene expression is not a straightforward effect. It is worth mentioning that IBD treatments, as anti-TNF drugs, could also influence the mRNA methylation levels and downstream expressions of the candidate. Thus, eQTL analyses in samples from treated patients would give further information about the involvement of m6A-SNPs in the regulation of candidate gene expression. Additional analysis of the potential effect of disease-associated m6A-SNPs in the candidate genes revealed that these variants are within RNA binding protein binding sites, so that the interaction of the mRNAs with their target proteins might be influenced by genotype and therefore their regulatory effect would be in an allele specific manner. Indeed, we observed that specific genotypes in disease-associated variants led to secondary structure changes in *UBE2L3* and *SNAPC4* mRNA molecules. Hence, it is important to mention that SNPs could indirectly affect gene expression via more complex mechanisms that were probably missed by an eQTL analysis. These results underline the importance of considering different layers of gene expression regulation when analyzing the potential functional effects of disease associated SNPs.

We used the TREW database to evaluate the m6A machinery proteins that have been found to bind our candidate genes. We observed that the YTHDF1 reader had the ability to interact with all candidate genes. Interestingly, YTHDF1 is implicated in the development of colorectal carcinomas, and its expression has been shown to be significantly augmented in this type of cancer [28]. Although the only changes in *YTHDF1* gene expression that we observed point to a downregulation of this gene in UC patients, the induction of this protein together with the augmented overall m6A methylation found in response to IFNγ suggest that a continuous inflammatory environment could lead to an increase in m6A and m6A regulatory proteins. Indeed, we observed that *WTAP* writer expression is upregulated in UC patients. WTAP increase has been also found in gastrointestinal tumors and its

expression has been related to T lymphocyte infiltration [29]. These results suggest that increased m6A methylation and the induction of m6A machinery proteins could predispose individuals with IBD to develop gastrointestinal malignancies. Interaction analysis also showed that *SLC22A4* and *SNAPC4* can bind different m6A writers, even if these two genes had contrary expression trends in the disease condition. The different locations of the m6A-SNPs in these two genes (intron vs. gene body) evidence the diverse effects of the m6A motifs, based on their location. Co-expression analysis of m6A machinery proteins with candidate genes did not give the same information as the TREW interaction data, stressing, once again, the complexity of m6A mediated gene interactions.

The induction of *METTL3*, *METTL14* and *YTHDF1* after IFNγ stimulation confirmed the implication of the m6A pathway in the inflammatory response in intestinal cells. The pathophysiological role of IFNγ in IBD has been mainly attributed to its effects on the intestinal epithelium [25]. m6A RNA methylation has been shown to play important roles in the interferon response to viral infections [16] but, to our knowledge, this is the first study showing an increase in the overall levels of m6A in the presence of IFNγ in intestinal cells. Although the overexpression of *METTL3* and the silencing of *YTHDF1* did not give very striking results, we could observe certain alterations in some downstream candidate genes. As previously mentioned, m6A regulated pathways are generally complex, so further studies in the localization or stability of the mRNA or even in the protein expression of the candidates will provide more specific information on the implication that m6A exerts into these genes.

To summarize, our study shows that the combination of different datasets can be very helpful for describing novel candidate genes involved in disease pathogenesis and pinpoints m6A as a mechanism that could be helpful to understand the etiology of complex diseases. Moreover, our data suggest that the m6A pathway may play a role in the development of IBD and, as previously proposed for DNA methylation [30], our work opens a new window to the development of novel therapeutic approaches based on the regulation of mRNA methylation.

## 4. Materials and Methods

### 4.1. Selection of Candidate Genes

IBD, CD and UC associated SNPs, located within m6A peaks (m6A-SNPs) were selected. For this purpose, International IBD Genetic Consortium GWAS results [19] were retrieved and variants with a *p*-value $< 1 \times 10^{-8}$ and INFO score $> 0.8$ were kept. On the other hand, m6A peaks found in HeLa and HepG2 epithelial cell lines available on MeT-DBv2 [31] were used. Overlapping positions of disease-associated SNPs and m6A peaks were calculated using the intersect method implemented in Bedtools (v2.29.2) [23]. The disease-associated SNPs located within m6A peaks present in both cell lines were considered as hits of interest and were selected for further analyses.

Then, differential expression analyses of genes harboring m6A-SNPs were performed using expression data of both IBD subtypes available in the Gene Expression Omnibus (GEO). For Crohn's disease, available read counts of GSE85499 [32] were used; values were normalized by RUVSeq [33] R package [34] using the less variable 5000 genes as reference genes; differential gene expression analysis was carried out using edgeR [24] R package. For Ulcerative colitis, normalized values from GSE105074 [35] were used and Mann–Whitney test/GLM (Generalized Linear Model) were applied using R language. Genes that were differentially expressed and harbored a disease subtype-associated m6A-SNP were selected for further studies.

### 4.2. Analysis of m6A-SNP Regulatory Capacity

*Cis*-eQTL analysis was carried using the GTEx eQTL Dashboard tool [36]. Colon tissue was used as target tissue as it is the affected tissue in both CD and UC subtypes.

POSTAR2 database [37] was used to search RNA binding proteins (RBP) that may have their binding site on an m6A-SNP location. "Variation" submodule in "RNA" module was used to evaluate candidate genes and associated SNPs.

RTH RNAsnp Web Server of the Centre for non-coding RNA in Technology and Health (RTH) [38] was used to predict SNP effects on local RNA secondary structure. The most common mRNA transcripts sequences were extracted from Ensembl Genome Browser [36]. The m6A-SNP position and the allele change were indicated in the input sequence and a comparison, based on global folding (mode 1) was selected.

### 4.3. Candidate Gene and m6A Machinery Protein Interaction Analysis

m6A machinery proteins, described to interact with our candidate gene mRNAs, were evaluated using the TREW database (Target of m6A Readers, Erasers and Writers database), available in metDB-V2.0. This database collects ParCLIP-seq and MeRIP-seq data from different studies for the m6A regulator and reader proteins, including FTO, KIAA1429, METTL14, METTL3, WTAP, HNRNPC, YTHDC1 and YTHDF1 [31]. The m6A related proteins interacting with our targets in both HeLa and HepG2 epithelial cell lines were selected.

### 4.4. Candidate Gene and m6A Machinery Protein Co-expression Analysis

Whole genome expression data of controls and patients from the CD and UC datasets [32,35] were used for co-expression analyses between candidate genes and all the m6A machinery proteins that were found to interact with the candidates. Pearson correlation coefficients and $p$ values were calculated using GraphPad. Correlation was considered significant when $p < 0.05$.

### 4.5. Cell Lines and Stimulations

Intestinal HCT116 (#91091005) cell line was purchased from Sigma-Aldrich (Poole, UK). Cells were cultured in DMEM (Lonza, Basel, Switzerland, #12-604F) supplemented with 10% FBS (Millipore, Burlington, MA, USA #S0115), 100 units/mL penicillin and 100 µg/mL streptomycin (Lonza, #17-602E). For IFNγ stimulation, HCT116 cells were treated with IFNγ (R&D Systems, Minneapolis, MN, USA, #285-IF-100/CF) at a final concentration of 200 U/mL for 4 h.

### 4.6. Overexpression and Silencing

For METTL3, the overexpression 250 ng of plasmid from Addgene (#53739) was transfected using X-Treme Gene HP DNA transfection reagent (Sigma-Aldrich, #6366546001). Cells were harvested 48h post-transfection.

For YTHDF1 silencing, 30 nM of two different siRNAs against YTHDF1 (IDT, # hs.Ri.YTHDF1.13.1 and hs.Ri.YTHDF1.13.2) or negative control siRNA (IDT # 51-01-14-01) were transfected using Lipofectamine RNAimax reagent (Thermo Fisher Scientific, Waltham, MA, USA). Cells were harvested 48 h post-transfection.

### 4.7. RNA and Protein Extractions

RNA was extracted using NucleoSpin RNA Kit (Macherey Nagel, Düren, Germany, #740984.50) and proteins were lysed in RIPA buffer (150 mM NaCl, 1.0% NP-40, 0.5% NaDeoxicholate, 0.1% SDS, 50 mM TrisHCl, 1 mM EDTA).

### 4.8. Dot Blot

300 ng of RNA were heated for 3 min and rapidly put into ice. RNA was then crosslinked into a nitrocellulose membrane using UV. The membrane was blocked using 5% non-fatty milk in 0.1% PBST (0.1% Tween in PBS) and was incubated overnight with an m6A antibody (1:200) (Abcam, Cambridge, UK, #ab151230) at 4 °C. After washing the membrane in 0.1% PBST, it was incubated with a secondary HRP conjugate anti-rabbit antibody (1:10,000) (Santa Cruz Biotechnology, Dallas, TX, USA, #sc-2357) and finally developed using Clarity Max ECL Substrate (BioRad, Hercules, CA, USA, #1705062).

*4.9. Gene Expression Analysis*

500–1000 ng of RNA was used for the retrotranscription reaction using iScript cDNA Synthesis Kit (BioRad, #1708890). qPCR was performed using iTaq SYBR Green Supermix (Bio-Rad, #1725124). Reactions were run in a BioRad CFX384 and melting curves were analyzed to ensure the amplification of a single product. All qPCR measurements were performed in duplicates and expression levels were analyzed using the $2^{-\Delta Ct}$ method. Specific primer pairs used for expression levels determination are listed in the Supplementary Table S2.

*4.10. Western Blot*

Laemmli buffer 6X (62 mM Tris-HCl, 100 mM dithiothreitol (DTT), 10% glycerol, 2% SDS, 0.2 mg/mL bromophenol blue, 2% 2-mercaptoethanol) was added to proteins extracts and denaturized at 95 °C for 10 min. Proteins were migrated on 10% SDS-PAGE gels. Following electrophoresis, proteins were transferred onto nitrocellulose membranes using a Transblot-Turbo Transfer System (BioRad) and blocked in 5% non-fatty milk diluted in TBST (20 mM Tris, 150 mM NaCl and 0.1% Tween 20) at room temperature for 1 h. The membranes were incubated overnight at 4 °C with primary antibodies for METTL3, YTHDF1 and GAPDH at a 1:1000 dilution. Immunoreactive bands were revealed using the Femto ECL Substrate after incubation with a horseradish peroxidase-conjugated anti-rabbit or anti-mouse secondary antibody (1:10,000 dilution in 2.5% non-fatty milk) for 1h at room temperature. The immunoreactive bands were detected using a Bio-Rad Molecular Imager ChemiDoc XRS (BioRad).

**Supplementary Materials:** The following are available online at http://www.mdpi.com/2075-4655/4/3/16/s1, Figure S1: Heatmaps of the differentially expressed genes harboring an m6A-SNP (A) CD and (B) UC. Genes selected for further analyses are marked with an asterisk, Figure S2: Expression analyses of (A) CD and (B) UC candidate genes after 4 h IFNγ treatment. Data are represented as the mean and standard error of four independent experiments, Figure S3: Expression analyses of (A) CD and (B) UC candidate genes after YTHDF1 silencing. Values are represented as the mean and standard error of three independent experiments, Table S1: Information about differentially expressed genes harboring m6A-SNPs. Selected candidate genes for UC are highlighted in blue and those for CD in green, Table S2: Specific primer pairs used for RT-QPCR expression analyses.

**Author Contributions:** Conceptualization, A.C.-R.; Data curation, A.O.-G. and K.G.-E.; Formal analysis, I.S., K.G.-E. and A.C.-R.; Funding acquisition, A.C.-R.; Methodology, M.S.-d., A.O.-G. and I.G.-M.; Software, I.G.-M. and K.G.-E.; Supervision, I.S. and A.C.-R.; Writing—original draft, M.S.-d., I.S. and A.C.-R.; Writing—review & editing, A.O.-G., I.G.-M. and K.G.-E. All authors have read and agreed to the published version of the manuscript.

**Funding:** This work was supported by Spanish Ministry of Science, Innovation and Universities (grant PGC2018-097573-A-I00) to A.C.-R. I.S. was funded by research project grant 2015111068 of the Basque Department of Health and a Research Grant from the European Foundation for the Study of Diabetes. M.S.-d. and I.G.-M. are predoctoral fellows funded by grants from the University of Basque Country and A.O.-G. is a predoctoral fellow funded by Basque Department of Education, Universities and Research.

**Conflicts of Interest:** The authors declare no conflict of interest.

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
