# Peer review of "Implication of m6A mRNA Methylation in Susceptibility to Inflammatory Bowel Disease"

_2075-4655, 2014_

Round 1
Reviewer 1 Report
it is an interesting in silico based analysis however will be interesting to be tested in human samples. The authors should include studies on IBD methylation ie Medicine (Baltimore). 2014 Dec;93(28):e309. and also should comment if there is a possibility the epigenome alteration studied to be altered by therapeutic approached used on iBd treatment
Author Response
As the reviewer comments testing these results in human samples from IBD patients will be very interesting and we hope to do it in future studies.
As suggested by the reviewer, we have mentioned studies about methylation already performed in the context of IBD (lines 64-68 and line 521) and have commented on the possibility of IBD treatment to alter the methylation levels (lines 459-472).
Reviewer 2 Report
The authors using online available GWAS, m6A and transcriptome data found candidate genes corresponding to ulcerative colitis and Crohn’s disease. In further experiments they found that such genes seem to be regulated in m6A dependent manner. The manuscripot is well written and discussed, however some technical amendments are necessary:
Line 25 Crohn’s not Chron’s
Line 150 and elsewhere - please check and be consistent in using dot comma as decimal separator and comma as thousands separator.
Fig1A) Manhattan plots should be replaced with better quality images.
Line 234, 236 and 346 m6A not m6a
Table 1 should be redesigned to improve clarity and readability.
Author Response
We thank the reviewer for his thorough evaluation, we have now corrected the mistakes he pointed.
As recommended by the reviewer we have replaced the Manhattan plots with better quality images and have redesigned Table 1.